# The Impact of Race and Gender-Related Discrimination on the Psychological Distress Experienced by Junior Doctors in the UK: A Qualitative Secondary Data Analysis

**DOI:** 10.3390/healthcare11060834

**Published:** 2023-03-12

**Authors:** Niha Mariam Hussain, Johanna Spiers, Farina Kobab, Ruth Riley

**Affiliations:** 1College of Medical and Dental Sciences, University of Birmingham, Birmingham B15 2SQ, UK; 2School of Health Sciences, University of Surrey, Surrey GU2 7XH, UK; 3Department of Social Work and Social Policy, School of Social Policy, University of Birmingham, Birmingham B15 2SQ, UK

**Keywords:** junior doctors, intern, residents, registrars, racial discrimination, gender-based discrimination, psychological distress, qualitative

## Abstract

Almost half of NHS doctors are junior doctors, while high proportions are women and/or Black, Asian, and Minority Ethnic (BAME) individuals. Discrimination against this population is associated with poorer career-related outcomes and unequal representation. We aimed to qualitatively explore junior doctors’ experience of workplace racial and gender-based discrimination, and its impact on their psychological distress (PD). In this study, we carried out a secondary analysis of data from a UK-based parent study about junior doctors’ working cultures and conditions. Interview data was examined using thematic analysis. Transcripts (*n* = 14) documenting experiences of race and/or gender-based discrimination were sampled and analysed from 21 in-depth interviews conducted with UK junior doctors. Four themes were generated about the experiences and perpetrators of discrimination, the psychological impact of discrimination, and organisational interventions that tackle discrimination. Discrimination in various forms was reported, from racially charged threats to subtle microaggressions. Participants experienced profoundly elevated levels of PD, feeling fearful, undermined, and under-confident. Discrimination is associated with elevated levels of PD, whilst negatively impacting workforce sustainability and retention. This reduces the opportunity for more diversity in NHS medical leadership. We encourage NHS hospitals to review their policies about discrimination and develop in-person workshops that focus on recognising, challenging, and reporting workplace discrimination.

## 1. Introduction

Approximately 131,815 doctors practice in the NHS, almost half of whom are junior doctors [1]. The NHS workforce is diverse; of all registered doctors, 48% are women [2] and 41.9% are Black, Asian, and Minority Ethnic (BAME) individuals [3]. Nonetheless, discrimination against these individuals is associated with poorer career-related outcomes and unequal representation across training grades [4]. Despite forming a larger proportion of all junior doctors, they remain under-represented at consultancy level [5,6]. Doctors are theoretically and legally protected by NHS policies and legislation, including the 2010 Equality Act [7]. However, recent British Medical Association (BMA) surveys found that discrimination was common, with 76% and 91% of doctors experiencing workplace racism and sexism in the previous two years, respectively [6,8]. 

Discrimination negatively affects junior doctors’ wellbeing [9] and may contribute to psychological distress (PD). Compared to other healthcare professionals, junior doctors are already more vulnerable to PD [10], with the 2021 General Medical Council (GMC) National Training Survey concluding that 44% of trainees felt distressed and emotionally exhausted to at least a “high” degree [11]. This figure warrants research into the impact of workplace factors, such as discrimination, on PD. 

This secondary data analysis (SDA) aims to qualitatively explore workplace racial/gender-based discrimination and junior doctors’ PD. Discrimination negatively impacts workforce sustainability and retention [12], which consequently impacts workforce wellbeing [13]. BAME and female doctors’ underrepresentation in leadership positions [3,6] limits NHS innovation due to a lack of different perspectives or ideas for improving care [14]. This SDA could, therefore, inform future interventions which tackle discrimination and promote diverse workforce cultures. 

There is a lack of studies on doctors’ PD. Relevant research was largely retrospective and quantitative, with studies focusing on individuals’ need to develop resilience [15], thus neglecting organisational sources of PD and potential initiatives targeting systemic factors. 

### 1.1. Psychological Distress

PD encompasses the non-specific symptoms of stress, burnout, anxiety, and depression [16], and ranges from subliminal levels of stress to acute suicidality [17]. High rates of PD in doctors were reported in both UK and international studies [10,11], with an Australian report concluding that doctors who experienced higher rates of PD were typically female and under 30 years old [18]. Riley et al. qualitatively explored the myriad workplace factors that junior doctors perceived contributed to their PD, with 66.7% identifying racial or gender-based discrimination as a source of PD [19]. It was agreed that further analysis was warranted: this study is an SDA of that project. The term ‘PD’ was used in this SDA to ensure that participants were not misrepresented between the projects.

The minority stress theory describes chronically elevated levels of PD in individuals from stigmatised minority groups [20]; discrimination is associated with increased levels of burnout and suicidality [21] and predisposes doctors to behaviours such as smoking and alcohol misuse [22]. 

### 1.2. Discrimination in the NHS 

Female doctors may experience discrimination in a variety of forms, ranging from subtle microaggressions to sexual harassment [23]. They may also defer personal life decisions and face additional barriers to professional development, further contributing to burnout [24]. Similarly, BAME doctors experience ‘disproportionate and excessive monitoring’ by security when entering hospitals and they also experience White provider preference from patients [25]. Those who have not yet been discriminated against may still experience anticipatory anxiety from pressure to perform better than their White, male counterparts [26].

Consultants were found to be the most cited perpetrators of discrimination [27]. The social dominance theory hypothesises that hierarchies, in this case determined by training grade, may facilitate some consultants to bully junior doctors [28], resulting in disempowering work environments, which negatively influence junior doctors when reporting discrimination. Other perpetrators included other women, typically nurses and midwives [29], who could feel marginalised in the NHS due to intra-sex competition [6,30]. Women may internalise patriarchal messages about being less competent than men [31]. Furthermore, the belief that there is ‘one seat at the table’ available for women [32,33] may lead to rivalry, manifesting as women mistreating, undermining, and isolating themselves from other women to increase their standing amongst men, whilst facilitating discrimination. 

### 1.3. Intersectionality

Intersectionality considers the relationship between social identities, such as race and gender, which create overlapping and interdependent systems of discrimination or disadvantage [34]. Ignoring intersectional experiences creates the potential for White women to trivialise, ignore, or deny racism experienced by BAME women or for BAME men to ignore sexism experienced by BAME women [35]. The absence of research pertaining to BAME women has ramifications on healthcare and signifies a failure to consider intersectional experiences, due to the ‘whitewashing’ of medical research [36]. For example, some BAME women reported that whitewashing mental illnesses resulted in them feeling ignored and alienated by society, whilst validating pre-existing mental illness stigmas [37]. An intersectional approach helps in understanding the impact of amplified discrimination faced by BAME female doctors, as culture and religion, alongside gender and experiences working as a doctor, can create greater sources of stress.

Currently, no other qualitative study explored the impact of racial or gender-based discrimination on PD experienced in the junior doctor population, with related studies rarely considering the intersectionality between ethnicity and gender [23,35]. This SDA aims to address these gaps. We conclude that junior doctors who experienced racial and/or gender-based discrimination also experienced profoundly elevated levels of PD, with some feeling fearful, undermined, and under-confident at work. Our participants identified discriminatory experiences as contributing factors to their elevated levels of PD, whilst appreciating the negative impact it also had on the junior doctor workforce’s sustainability and retention.

## 2. Materials and Methods

### 2.1. Study Design

The present SDA obtained interview data from an England-wide study conducted by Riley et al. [19]. Fourteen of the twenty-one transcripts included in the parent study directly reported doctors’ experiences of racial or gender discrimination. Qualitative methods were employed, as they allowed a dynamic exploration of perceptions about discrimination and PD. Participants were asked about their experiences of discrimination, although some also discussed it in response to open-ended questions about sources of PD. The parent study’s topic guide [19] allowed for unanticipated sources of PD to be identified but did not explicitly enquire about discrimination. Whilst this limited this SDA, the topic guide was examined to contextualise the revisited transcripts. 

### 2.2. Recruitment 

A detailed account of recruitment and sampling can be found in the parent study’s methodology [19]. Participants were recruited by Riley et al. via purposive sampling, with the aim of capturing diverse characteristics, including gender, age, ethnicity, sexuality, and disclosure of either psychiatric diagnoses or previous suicidality [19]. This allowed the results to be transferable to junior doctors working in NHS England hospitals, but limited generalisability to doctors working abroad, given differences in health systems and culture. Those currently experiencing acute severe mental illnesses or suicidality were excluded to avoid causing additional psychological harm. A participant demographic table summarising the participants included in this SDA is detailed in Table 1. 

### 2.3. Interviewing 

A detailed account of the interviewing process can be found in the parent study [19]; interviews were conducted between November 2019 and May 2020 and took place both face-to-face and virtually. They ranged from 29 to 102 min in length (mean = 62.8 min). Pre-interview, participants were reminded that their data would be used on subsequent research projects that the team wished to undertake, including any findings that deserved further exploration. Interviews were transcribed verbatim post-interview; for this SDA, anonymised transcripts were stored on a password-protected, encrypted folder on a secure university server. Transcripts were then uploaded to NVivo 12 [38], where they were stored as a password-protected project. 

### 2.4. Analysis 

Braun and Clarke’s thematic analysis was employed, as it assists researchers in systematically examining perspectives [39]. Elements of narrative analysis were also used to analyse the dataset, given that the focus of this research was experience-centred. 

First, all transcripts involved in the parent study were read to confirm which were relevant to the SDA. The 14 relevant transcripts were re-read to assist familiarisation and reflection on potential codes. NVivo 12 was used to help manage the large volume of transcripts. 

Analysis was conducted by a multidisciplinary team consisting of a medical student (NMH) and three PhD social scientists (RR, JS, and FK). Analyst triangulation was undertaken to improve the study’s rigour [40,41]. Two investigators (RR and JS) independently double coded the three richest transcripts. Codes were then compared against the lead investigator’s codes to discuss any differences. Negative case analysis was undertaken to explore findings that appeared to contradict emerging patterns. Presenting alternative viewpoints improved credibility and assisted in broadening and confirming emerging themes [42]. A list of codes, derived from the data, was created to simplify the reviewing process, where Saldana’s two cycles of coding were employed [43]. This condensed 101 initial codes into 18 complex ones. A thematic mind-map helped to visualise relationships between codes and arrange them into potential themes; this step was repeated following discussions within the research team until the best fit was achieved. The final list of themes is summarised in Table 2.

### 2.5. Reflexivity

The lead investigator of this SDA identified as a BAME, female medical student who experienced discrimination, and, therefore, had personal and professional interest in the topic. Her identity also provided experiential insight which non-BAME researchers may have overlooked when undertaking the parent project. Analysis was performed through an intersectional lens, whereby the investigator’s research paradigm was broadly emancipatory and allied to both critical feminist and race theories [44,45]. 

## 3. Results

A summary of the included themes is detailed in Table 2. Included participant quotations were edited to improve readability.

### 3.1. Theme 1: Experiences of Discrimination

Whilst all included participants experienced workplace racism and/or sexism, the nature of experiences ranged from harmful stereotyping to aggressive threats and harassment. British BAME doctors experienced racial microaggressions via questions from others about where they were *“really” **[P1]*** from, after they had already stated that they were born in England. This made them feel defensive in justifying the origins of their identity. Others received patronising comments from surprised patients about their English language proficiency, despite being educated in the same cities as their White colleagues. One participant complained that her peers refused to attempt pronouncing her name, given the perceived difficulty in pronouncing it:
*“The number of times I’ve had people say, ‘Oh, I’m not going to try and pronounce that.’ If I introduced myself as Dr. Smith, they would listen straightaway and then they would say to the next person, ‘I’m seeing Dr. Smith,’ whereas with me, it’s quite different.”***P11**

Female doctors expressed frustration over sexist stereotyping, which resulted in them disproportionately being mistaken for nurses or secretaries. When considering entering competitive specialties, such as surgery, females also received more comments from colleagues expressing concern or disapproval, making them more hesitant:
*“Are they telling me that they wouldn’t choose surgery for themselves, or are they telling me not to apply because I’m a woman? Would they tell the same thing to a man?”***P5**

Whilst some female participants did not explicitly report being harassed, they were *“pestered”* [P8] by their male colleagues, who sometimes made inappropriate sexual comments at work, masquerading as jokes:
*“Male consultants are inappropriate towards female colleagues […] either not controlling where their eyes are going or when we have a meeting, making a joke about it being a date.”***P10**

Some BAME participants also perceived that their institution discriminated against them, given that they (on average) faced more complaints and investigations by the GMC than their White counterparts. One participant referenced the Bawa-Garba case [46] to exemplify how the fear of facing investigations affected doctors:
*“What happened to [Dr. Bawa-Garba] was because of her colour and her religion, possibly because of her gender; you just can’t be too careful nowadays, I think that has a profound effect on the way we work, and our stress levels.”***P1**

Some perceived that ethnicity contributed to a lack of parity in opportunities to progress to senior clinical posts; a participant recognised this whilst talking about his colleague, a Black, international medical graduate (IMG):
*“They’ve never given him a permanent job anywhere […] he’s been a locum consultant for years. There are people in the same circumstances: their English is not as good or they look a certain way, so they’re not taken seriously, and I think that’s a shame.”***P3**

#### Intersectional Experiences

Some 28% of included participants were BAME women, who experienced heightened levels of discrimination compared to White women and BAME men. One participant described how her identity often left her feeling nervous (about anticipating discrimination), shamed, and powerless following incidents involving discrimination with patients:
*“We’d walk away with our tail between our legs and then get somebody else to see the patient, thereby giving the racist or sexist patient what they want. I am a female and an ethnic minority, and we feel burdened more than ethnic males or the White females.”***P11**

These participants also felt side-lined and undermined by all men, including those from BAME backgrounds; one participant recalled witnessing her female senior be ignored by male doctors, despite her seniority and expertise:
*“I was with an Asian, female doctor, who was the senior registrar on call. We were seeing a patient in A&E with loads of White male doctors—one Asian male as well. The patient was under our care, but as they got sicker, they all took over when actually, my senior was perfectly competent.”***P14**

Feeling more vulnerable and overwhelmed by having to defend their identity from both racism and sexism made BAME women less likely to challenge poor behaviours they observed in hospitals:
*“There’s this constant, ‘you’ve got to pick your battles.’ I can’t try and be the spokesperson for everything that’s going on.”***P2**

One participant believed that her identity would prevent her complaints being taken seriously: she recounted wanting to raise a concern about the poor treatment of junior doctors during an orthopaedic rotation. Whilst she was unable to do so, an Asian male junior doctor raised this concern for her.
*“He was in that position [to raise a concern], socially, due to his appearance. He had like this big, massive beard, and a really deep voice…I just don’t think I would have gotten away with the flippant complaints that he made.”***P13**

This speculates that her colleague’s identity as a male was advantageous to him being able to raise concerns whilst she was unable to, despite being the same ethnicity and in a similar situation on the same ward as him.

### 3.2. Theme 2: Perpetrators of Discrimination

Perpetrators included patients, senior doctors, and midwives. Patients sometimes coupled offensive remarks with aggressive behaviours; one participant was verbally abused after he referred a patient’s child to social services:
*“[The patient] launched into an aggressive racist tirade and threatened to kill me. He called me a dirty P***, and said he was going to murder me, he was going to hunt me down”***P1**

The patient then demanded to see a ‘proper doctor;’ clarifying that he meant a White doctor. Similarly, others mentioned that patients often expressed ‘White-provider preference’ during racist encounters:
*“People are more sceptical of ethnic people. A patient once said [to a participant’s colleague] ‘I don’t want to be treated by anybody in a headscarf,’ and then made a complaint against her.”***P7**

Gender-based discrimination perpetuated by fellow staff was common in Obstetrics and Gynaecology, despite it being a female-dominated field [47]. Here, perpetrators were typically other females, notably members of the midwifery team, who were more likely to undermine a female doctor’s management plans:
*“I don’t know if female doctors are seen as more of a threat in a female-dominated area, but you’re more likely to have somebody argue against or b*tch about you; you can make the same plan as a male doctor and to be ignored by other women is very frustrating.”***P10**

Participants perceived that midwives *“warmed up quicker to the male doctors”* [P14], whilst mistrusting female colleagues. Females, therefore, had to work harder than males to integrate with the wider team during a rotation.

Similarly, participants acknowledged a power imbalance between perpetrators and ‘victims’ of discrimination, where the former were usually hierarchically superior to the latter. A female participant recalled that she was regularly asked to complete excessive secretarial tasks by male seniors, which she struggled to refuse. This may have been gender-associated, as the participant discussed this alongside her experiences of sexism:
*“I found it very difficult to say no [to consultants], because you need them as a reference to come back to work there… My consultant used me as more of a personal assistant. He’d say, ‘Oh, can you run to this building and get my Dictaphone?’ That’s not right, is it?”***P8**

This hierarchy, coupled alongside bullying and poor work cultures, may create unsettling environments for junior doctors and influence the extent of their PD, as well as influence help-seeking or reporting behaviours. 

#### Denial of Racism

Whilst all participants acknowledged that gender discrimination was common, an unconventional narrative found that some participants denied the existence of racism in the NHS. These participants suggested that it was increasingly harder to perpetuate racism at work, given the diversity in ethnicities amongst NHS doctors.
*“You can’t really have a racial bias anymore. Even if you secretly have it, you can’t express it because there are so many different ethnicities of people working.”***P11**

These participants believed that since racism is rare, it is not a workplace source of PD:
*“I don’t think that racism is prevalent in the NHS…I have heard the odd story, but very, very rarely.” ***P6**
*“People were super accepting of me, as they were of others who were from different areas of the world… I think the UK is incredibly accepting and welcoming.” ***P3**

Negative case analysis revealed that those who either expressed positive feelings about the way their ethnic identity was perceived or suggested that racism is not an issue in the NHS were typically White and worked in White-majority regions of the UK. This narrative contradicted the existing literature (which suggested that racism is common in NHS settings) and will be further explored in the discussion.

### 3.3. Theme 3: Psychological Impact of Discrimination

The psychological impact of discrimination may be broken down into two subthemes. Participants detailed both experiences pertaining to the ‘symptoms’ of PD and feelings that were not symptoms of PD. Rather, these latter feelings led to the internalisation of discrimination and were antecedents of PD.

#### 3.3.1. Internalisation of Discrimination

Some of the feelings participants described suggested the internalisation of discrimination; workplace sexism left females feeling less confident than their male counterparts, which manifested as underselling their clinical abilities, being less likely to volunteer opinions during group discussions, and not being appropriately assertive at work. One participant summarised:
*“Male colleagues appear more confident when they are no more competent…I think males are able to say, ‘yeah, of course I can do this, I’ve done three before.’ I think as a woman you’re more likely to be cautious.”***P4**

Additionally, females doubted their abilities as doctors, with some expressing sentiments in keeping with impostor syndrome. Women seemed less likely to respond to invitations to lead committees because they *“don’t feel deserving to be called the expert,”* [P4] and felt less able than men in their field. Discrimination, when perpetuated by seniors and accompanied by hostile behaviours, facilitated some participants’ internalisation of comments about their incompetence:
*“When consultants have been standoffish, you feel an awful lot more pressure and you’d be terrified of things going wrong...and definitely less likely to speak up or express a vulnerability [about not doing a good job]”***P9**

Some participants also speculated that senior female doctors often overcompensated for their under confidence by projecting themselves as hypermasculine or overly tough. Participants offered examples of their female colleagues deliberately seeking out high-adrenaline, emotive, emergency situations to try and demonstrate their importance, knowledge, or leadership abilities.
*“I worry that some of us end up overcompensating to be more manly or assertive… sometimes the female bosses end up being even harsher and stricter [than males]”***P13**

Participants felt underlying stress knowing that BAME doctors were more likely to be investigated by the GMC, recounting that they were not fully comfortable working, as they had to remain *“hyper-vigilant”* [p7] to avoid referrals:
*“It makes me feel more anxious and try to cover all angles and try not to get complaints. Rather than thinking about treating the patient, I’m more worried about projecting myself… I’m constantly watching my back” ***P7**

Discrimination, alongside generic bullying from seniors, left participants feeling scared to interact with other members of their team; one participant who was bullied by seniors recounted having to tell her consultant that she was too sick to work the next day. She recalled:
*“I was sat there for the rest of the day and instead of recovering, I was absolutely petrified of what people at work would be saying or thinking about me because I couldn’t get out of bed that day”***P6**

Individuals implied that they felt powerless and believed that discrimination was *“a rite of passage,”* [P9] for minority doctors. Whilst they are not symptoms themselves, lack of confidence, fear, and feeling undermined all contribute to the internalisation of discrimination, which causes PD.

#### 3.3.2. Symptoms of Psychological Distress

Participants reported low moods and the non-specific symptoms of PD whilst recounting past incidents involving discrimination. Whilst this was consistently implied in the above quotations, one participant explicitly summarised his psychological state following an incident:
*“On occasion there’s been racism and actually that was one of the most stressful things I’ve ever had…I’d only just graduated a year prior to it and at that point that was extremely upsetting.”***P1**

Many participants experienced anticipatory anxiety, as they believed that they would experience discrimination in their careers, even if they had not previously experienced it. Consequently, participants distrusted bodies such as the GMC, and felt nervous around other common perpetrators:
*“The system is out to catch you…you have to be on your guard.”***P7**
*“You watch nurses employ a very defensive culture, so that makes you feel on edge. You feel like you should be defending yourself too.”***P8**

Finally, one participant concurred with a reflection offered by an interviewer, who suggested that repeated microaggressions result in individuals experiencing constant, subliminal stress, coupled with anticipatory anxiety:
*“It’s the little nuances that kind of chip away…they’re [microaggressions] not going to get you fired, but they’re the ones that you carry the burden and the stress of in your mind because it’s not big enough to sort out.”***P3**

### 3.4. Theme 4: Tackling Workplace Discrimination

This theme details participants’ perceptions about challenging discrimination either individually or systemically. Participants believed that the largest barrier to challenging discrimination was that offences were subtle, “*fleeting comments*” [P2]. This, plus a lack of evidence, hindered individuals from reporting incidents. Sometimes, perpetrators were seniors who oversaw and influenced training, making participants reluctant to challenge discrimination, fearing that the senior may hinder their progression out of spite. One participant recalled that a colleague asked for sick leave to overcome severe morning sickness. Her consultant refused her leave, stating the following:
*“He said, ‘No, you chose to be pregnant,’ and he refused to let her leave. I think she complained, but you can’t do anything because when that’s the clinical lead, what can you do?”***P8**

Participants tackled discrimination individually by reporting and confronting perpetrators. Whilst few reported positive outcomes, others reported that bullying intensified after confronting perpetrators. A further barrier to reporting discrimination was the impact that it had on participants’ experience of PD; participants experienced symptoms of anxiety and burnout from the prospect of raising concerns:
*“I don’t have the energy to report this. I’m physically and mentally exhausted. Can I really be bothered with the headache this is going to create?”***P11**

Participants highlighted the need to better signpost both counselling and reporting services for doctors experiencing discrimination and suggested further training to raise awareness about discrimination. Additionally, a policy that would allow doctors to refuse non-critical care to patients who perpetuated discrimination could tackle discrimination. One participant explained:
*“We’ve got to be able to say, ‘what you said is unacceptable, you are no longer allowed on these premises;’ there’s no way, as far as I’m aware, of denying [perpetrators] the right to come to your practice.”***P1**

The aforementioned policy was enforced in April 2020 [48]; however, it is likely that this will need to be better advertised, so that doctors are aware of their newfound right. 

## 4. Discussion 

### 4.1. Key Findings

This SDA qualitatively presents an in-depth insight into the psychological impact of workplace discrimination from both BAME and female perspectives. Although participants did not report specific symptoms of mental health problems such as anxiety or stress, our findings indicate the junior doctors in this study understood their experiences of both racial and gender-based discrimination to have impacted their perceived levels of PD. Participants also felt undermined, fearful of interacting with others, and lacked confidence at work. The minority stress theory suggests that these feelings lead to the internalisation of discrimination, which causes PD [20,49]. Furthermore, discrimination evoked feelings associated with impostor syndrome, which was found to predispose individuals to PD and is associated with impaired job satisfaction and performance, whilst contributing to the development of depression and anxiety [50].

The wide range of organisational and societal barriers to reporting incidents evoked further negative emotional responses, which further contributed to PD. Even without external barriers, the internalisation of discrimination often leads to impaired cognitive function, planning, and decision making [51], which may consequently hinder doctors’ abilities to seek interpersonal support following incidents. 

Another key finding was that some participants did not consider racism a serious issue in the NHS. Negative case analysis explained that these ethnocentric perspectives came from participants who were typically White. Such participants may have possessed ‘White privilege [52] ’, which allowed them to interact with others without having to worry about how their race may be perceived. Thus, these individuals may have been less likely to recognise and challenge subtle forms of racism, but instead more likely to trivialise or perpetuate it themselves.

This argument was exemplified by findings from a recent BMA ‘Racism in Medicine’ survey, which found that over 90% of BAME doctors and medical students believed that racism in medicine is an issue, compared with only 64% of White doctors [23]. Similarly, an American survey found that the least frequently reported workplace problems included gender and racial-minority discrimination; however, respondents were mostly male (66.20%) and predominantly White (79.10%) [53]. This disparity was likely due to the dichotomies created between White and BAME individuals in defining and recognising discrimination. Furthermore, the typical inclusion of only White participants in psychological studies (an example of whitewashing) may contribute to discriminatory experiences being silenced or underreported in research [54], further validating perceptions that workplace racism is rare and ensuring that the problem remains unchallenged. Our results exemplified the need for interventions focusing on recognising and challenging discrimination in the NHS. 

### 4.2. Study Strengths and Limitations

SDAs can be easy to conduct, requiring little expenditure and allowing researchers to generate new insights and perspectives on existing research. However, the dataset used in this SDA was used to answer a different research question from that of the parent study [19]. Since we were unable to return to participants, the gaps in analysis were answered via an iterative literature search. Whilst small sample sizes are typical (and effective) in qualitative research, these findings were limited by the under-representation of certain ethnic groups in the parent study, notably Black-African and Black-Caribbean groups. Whilst both the present and parent study included BAME participants, a ‘one-size-fits-all’ approach to discrimination fails to grasp the differences in its extent and impact between different ethnicities [55]. Different ethnicities have differing workplace experiences and, thus, would experience PD differently. A future primary qualitative study focusing on this topic should be performed; to address the problem of underrepresentation, future studies should include participants from a wider range of ethnicities and distinguish between the experiences of British BAME doctors and IMGs.

Interviews were conducted by two female researchers (JS and FK), including one BAME woman, who interviewed all the BAME participants. This was advantageous, as those reporting discrimination may have felt more comfortable to give an in-depth and honest account, perceiving that their experiences were better understood by the interviewers [56]. The lead investigator was also able to collaborate with the parent study’s research team and was, therefore, aware of any study-specific nuances in the data collection that may have otherwise influenced the analysis.

### 4.3. Recommendations

Based on the above findings, we make two recommendations. Firstly, that NHS trusts design and deliver mandatory in-person workshops to raise awareness and educate all healthcare staff about recognising, challenging, and reporting discrimination, even when subtly presented. This would replace the virtual training staff typically complete at home [57]. When staff complete ‘equality, diversity, and inclusion’ training at home, they may be unable to discuss training content or common questions with others, nor can hospitals guarantee that staff genuinely engaged with the teaching material. In-person workshops could allow educators to tailor teaching material so that it covers information about NHS trust-specific reporting procedures or accessing local counselling services. This would empower doctors to act if they experience or observe workplace discrimination.

Our second recommendation is that NHS trusts ensure doctors are aware of the new NHS policy which states that doctors can refuse to give non-critical care to patients who perpetrate discrimination, and refer such patients to other physicians [48]. The policy, which was enforced in 2020, acknowledged that UK healthcare professionals, including doctors, are protected by laws such as the Equality Act, and that they have a right to work in an environment which is free from discrimination. Nonetheless, we reiterate that this policy would only be implemented in a non-critical care setting, and that whilst a doctor may refuse to treat patients that perpetrate discrimination towards them, it is the doctor’s responsibility to ensure that they are referred to someone who is just as competent to treat the patient, and that this decision is communicated to the patient in a respectful manner. This ensures that doctors are protected and that the care of patients is not compromised, as they would still be able to receive treatment elsewhere.

## 5. Conclusions

This SDA found that both racial and gender-based discrimination were synonymous with profoundly elevated levels of PD in junior doctors. Doctors also reported feeling fearful, underconfident, and undermined following experiences of discrimination; when chronically manifested, these are antecedents to PD and psychiatric conditions such as depression and anxiety. This SDA also identified a plethora of individual, interpersonal, organisational, and societal barriers faced by individuals who wished to challenge discrimination. This further contributed to doctors’ PD.

These findings have implications for medical leadership: whilst women and BAME doctors are key constituents of the medical workforce, push factors such as discrimination negatively impact workforce sustainability and retention, reducing the opportunity for diverse leadership. Nonetheless, cautious optimism about the improving treatment of doctors from minority groups could be dependent on interventions that tackle discrimination being consistently implemented across hospitals.

## Figures and Tables

**Table 1 healthcare-11-00834-t001:** Participant characteristics.

Total Number of Participants Included in Analysis (n)	14
Gender: Female (n, %)	11 (78.6%)
Ages: (n, %)	
20–29 years (n)	7 (50%)
30–39 years (n)	7 (50%)
Current Speciality (n, %)	
General Practice	2 (14.3%)
Paediatrics	1 (7.14%)
Obstetrics and Gynaecology	5 (35.7%)
Medicine	3 (21.4%)
Accident and Emergency	3 (21.4%)
Ethnicity (n, %)	
White	8 (57.1%)
South Asian	4 (28.6%)
Chinese	1 (7.14%)
Asian ‘Other’	1 (7.14%)
Psychological Assessment; participants reported having the following (either at baseline or in the past) (n, %)	
Stress	13 (92.9%)
Anxiety diagnosis	11 (78.6%)
Depression diagnosis	6 (42.9%)
History of Suicidality or Self-Harm	3 (21.4%)

**Table 2 healthcare-11-00834-t002:** A summary of final themes, sub-themes, and complex codes.

**Theme 1**	**Experiences of Discrimination**	
**Subthemes**	**Types of discrimination**	**Intersectional experiences**
**Complex codes included**	Types of racial discrimination Types of gender discrimination Institutional discrimination	Experiences of BAME women
**Theme 2**	**Perpetrators of discrimination**	
**Subthemes**	**Perpetrators**	**Denial of racism**
**Complex codes included**	Discrimination at medical school Perpetrators of discrimination Specialities notorious for discrimination	Belief that the NHS is not racist anymore
**Theme 3**	**How discrimination makes doctors feel**	
**Subthemes**	**Internalisation of discrimination**	**Symptoms of ‘psychological distress’**
**Complex codes included**	Lack of confidence Fear caused by discrimination Feeling ignored, undermined or isolated because of discrimination Relationship between discrimination and other workplace pressures	Feeling anxious and upset after incidents. Feeling helpless and putting up with discrimination Anticipatory anxiety
**Theme 4**	**Handling workplace discrimination**	
**Complex codes** **included**	Tackling discrimination on an individual level Organisational ways to tackle discrimination Issues with raising concerns	

## Data Availability

This study was a secondary data analysis. The parent study did not receive ethical approval to share confidential data with any third party other than the study research team.

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
