# Peer review of "The Impact of Race and Gender-Related Discrimination on the Psychological Distress Experienced by Junior Doctors in the UK: A Qualitative Secondary Data Analysis"

_healthcare, 2023, doi:10.3390/healthcare11060834_

Round 1

Reviewer 1 Report

The topic that the authors of this manuscript have explored is extremely  important and concerns everyday professional functioning and experiencing  the impact of race and gender-related discrimination on the  psychological distress experienced by junior doctors in the UK. The authors are requested to take into consideration reviewer's recommendations and give response to each of them:

·       Line 25-16 (quote) '' We recommend excluding patients who perpetuate discrimination from non-critical care and developing compulsory in-person workshops on recognizing, challenging, and reporting workplace discrimination .” The quoted phrase must be rephrased because it sounds as a full of aggression and retaliation. It seems more  proper  to be used information phrase line 37-38 (quote) ,, Doctors are theoretically protected by NHS policies and legislation, including the 2010 Equality Act “ ,and thus it is recommended to raise issue of necessity re-evaluate NHS policies and legislation, including the 2010 Equality Act and modify such legislation for more effective protection  especially women, Black, Asian and Minority Ethnic (BAME) junior doctors in the UK .

·       The results of a study conducted on a group of less than 100 do not seems to be entirely reliable, unless current study are preliminary one and the same study will be conducted again on a group of at least 100 people in each of the study groups. If this is the case, the title of this research is recommended to  be modified and the phrase "Preliminary research" should be entered in the title.

·       Line 453-455 (quote) ,, Secondly, that NHS trusts ensure doctors are aware of the new NHS policy which  states that doctors can refuse to give non-critical care to patients who have perpetuated discrimination ''.  The quoted is recommended to be rephrased and adopted in example such as ,,  Secondly, that NHS trusts ensure doctors are aware of the new NHS policy which  states that doctors can refuse to give non-critical care to patients who have perpetuated discrimination upon refer such patients to other physicians”.

·       The method of referring to the references list in the main text of the manuscript does not meet the requirements of the journal at all

·       It is recommended to establish the paragraph "Conclusions".

·       The list of references in its current form does not meet the requirements of the journal. Revision is mandatory.

Author Response

Hello,

Thank you for reviewing our paper. We are glad that you can appreciate the importance of this topic and how it would fit into the scope of Healthcare’s publications. Your feedback was very useful to us and we have been able to make some changes that will hopefully further strengthen this piece. We would appreciate a second glance at our resubmitted work, as we are keen to have this published as a strong piece of work. To address your changes below:

  1. The minor changes regarding changing the wording of points (1) and (3) have been completed.
  2. Given the lack of words we had available in our abstract, line 25 now reads: “We encourage NHS hospitals to review their policies about discrimination and develop in-person workshops for staff that focus on recognising, challenging, and reporting workplace discrimination.”
  3. Line 461 now reads: “Our second recommendation is that NHS trusts ensure doctors are aware of the new NHS policy which states that doctors can refuse to give non-critical care to patients who have perpetuated discrimination, and refer such patients to other physicians.”
  4. Whilst this was not a part of your immediate recommendations, I’ve made a couple of changes to the recommendations section of this paper.
  5. Upon the recommendations of our other reviewer, we have reformatted Table 2 to make sure that it is easier to read
  6. We have added in a conclusion paragraph after the recommendations section.
  7. Both in-text references and the overall reference list has been changed to match the requirements of the journal.

We were a little confused about one of your comments: “The results of a study conducted on a group of less than 100 do not seems to be entirely reliable, unless current study are preliminary one and the same study will be conducted again on a group of at least 100 people in each of the study groups. If this is the case, the title of this research is recommended to  be modified and the phrase "Preliminary research" should be entered in the title.”

As you may know, our study is a qualitative one, rather than quantitiative. It is typical to have low sample sizes in such research, as it facilitates us as researchers to go in depth into the analysis of each participant’s interview transcripts. Additionally, it is rare to have a qualitative study that has over 100 participants, as data saturation would have been achieved much sooner than this- to recruit more participants to the study would be unnecessary (and potentially unethical). As our paper is a secondary data analysis, we would hope that a future primary study will be conducted, but this would not necessarily be with a larger sample size. With this in mind, we did not change the title of our work; I hope this is okay- please get in touch if you have any questions about this or anything else.

Thank you again for all of your support in reviewing our work. We look forward to hearing from you regarding any further recommendations.

Reviewer 2 Report

Thank you for the opportunity to review this manuscript. I hope my feedback below is helpful.

lines 66-67: there is an extraneous “and” in the sentence (“It was agreed that further analysis and was warranted:”)

line 91: Is “Intersectionality” meant to be (italicized): 1.3. Intersectionality ?

line 95: I believe the sentence is missing an “or”, as in: “…ignore or deny racism experienced by BAME women or for BAME men to ignore sexism experienced by BAME women”

line 109-110: The language of “association” gives me pause given this is a qualitative rather than quantitative study. Perhaps consider something like: “Participants identify discrimination experiences as contributing to elevated levels of PD whilst negatively impacting the sustainability of a career in medicine and therefore diminishing their intention to continue practicing" if your data support that, etc.

Table 2 - may want to check that the proof is displaying this table the way you wanted; looks like perhaps the text with bullets got centered rather than left-aligned, so none of the bullets line up - but maybe that was intentional.

lines 394-395: same comment as above — this language feels quantitative and longitudinal (“Given that participants reported non-specific symptoms of anxiety, stress, depression, and burnout following incidents involving discrimination, the findings demonstrated that both racial and gender-based discrimination led to elevated levels of PD”). Consider altering this sentence to read: “…the findings indicate these physicians understood their experiences of both racial and gender-based discrimination to have impacted their levels of PD” or something of that nature.

Author Response

Hello,

Thank you for taking the time to review our work, we found your comments to be very helpful, and have incorporated the following changes to improve the strength of this submission. To address some of your comments:

  1. We have made changes to formatting, spelling and grammar on the following lines, so that there are no mistakes in them: Lines 66-67, line 91, line 95.
  2. Line 109-110 now reads: “We conclude that junior doctors who experienced racial and/or gender-based discrim-ination also experienced profoundly elevated levels of PD…”
  3. Line 394-395 now reads: “Given that participants reported non-specific symptoms of anxiety, stress, depression, and burnout following incidents involving discrimination, the findings demonstrated that in this group, both racial and gender-based discrimination led to elevated levels of PD.”
  4. I have now changed the formatting of Table 2 so that the bullet points are better aligned.

I have also made some other changes, at the request of another peer-reviewer. These include:

  • Both the In text references and the reference list have been changed to match the requirements of this journal. The references are all in the correct order now.
  • A conclusion paragraph has been added
  • Some more information has been added to the recommendations section of the paper.

Thank you again for all of your support in reviewing our work. We look forward to hearing from you regarding any further recommendations.

Round 2

Reviewer 1 Report

The authors took into consideration  the reviewer's recommendations, but not all such recommendations  were used to improve the quality of the text. However, the most important issue seems to be the legal conflict that may arise in everyday medical practice in correspondence  phrase of revised first time manuscript , line 465-467 (quote) ,,  Our second recommendation is that NHS trusts ensure doctors are aware of the new NHS policy which states that doctors can refuse to give non-critical care to patients who  perpetrator discrimination, and refer such patients to other physicians [48]."  The more proper and ethical seems (first)  to notify patient about referral to other physician, (second)  and  gently advise patient  that  staff on duty is unable to give non-critical care to such patient. Otherwise,   this may cause legal conflict and threaten lawsuits against doctors. It should be remembered that physicians are bound by a code of ethics towards patients, regardless of whether the patient has discriminatory tendencies towards physicians or not. Therefore, aggressive language or intentional actions towards such patients are prohibited as  not ethical and may result in legal consequences.

Author Response

Thank you for getting back to us so quickly with your comments and ideas for revisions. We appreciate your concerns with our recommendations. We would like to reiterate that the recommendation we have proposed is that doctors are simply aware of their rights under this NHS policy. The policy acknowledges that doctors, are protected by law and that they have a right to work in an environment which is free from discrimination and that race/ethnicity/sex are protected characteristics.

To address the concerns, we decided to elaborate on the specifics of this policy, and have highlighted that the policy may only be implemented in non-critical care settings, and that the patient must be told about a decision to refer them to another doctor. We have added the following lines to our recommendation section:

“The policy, which was enforced in 2020, acknowledges that UK healthcare professionals, including doctors, are protected by laws such as the Equality Act, and that they have a right to work in an environment which is free from discrimination. Nonetheless, we reiterate that this policy would only be implemented in a non-critical care setting, and that whilst a doctor may refuse to treat patients that perpetrate discrimination towards them, it is the doctor’s responsibility to ensure that they are referred to someone who is just as competent to treat the patient, and that this decision is communicated to the patient in a respectful manner. This ensures that doctors are protected and that the care of patients is not compromised, as they would still be able to receive treatment else-where.”

We hope that our recommendation no longer seems aggressive as we have toned down the language and explained the policy further. 

Thank you again for supporting us with our work. 

Best wishes, 

Niha